# Effects of Follicle-Stimulating Hormone on Human Sperm Motility In Vitro

**DOI:** 10.3390/ijms24076536

**Published:** 2023-03-31

**Authors:** Rossella Cannarella, Francesca Mancuso, Nunziata Barone, Iva Arato, Cinzia Lilli, Catia Bellucci, Marco Musmeci, Giovanni Luca, Sandro La Vignera, Rosita A. Condorelli, Aldo E. Calogero

**Affiliations:** 1Department of Clinical and Experimental Medicine, University of Catania, 95123 Catania, Italy; 2Department of Medicine and Surgery, University of Perugia, 06123 Perugia, Italy

**Keywords:** FSHR, sperm motility, spermatozoa, FSH

## Abstract

To evaluate whether the follicle-stimulating hormone (FSH) receptor (FSHR) is expressed in human spermatozoa and the effects of FSH incubation on sperm function. Twenty-four Caucasian men were recruited. Thirteen patients had asthenozoospermia, and the remaining 11 had normal sperm parameters (controls). After confirming FSHR expression, spermatozoa from patients and controls were incubated with increasing concentrations of human purified FSH (hpFSH) to reassess FSHR expression and localization and to evaluate progressive and total sperm motility, the mitochondrial membrane potential, and protein kinase B (AKT) 473 and 308 phosphorylation. FSHR is expressed in the post-acrosomal segment, neck, midpiece, and tail of human spermatozoa. Its localization does not differ between patients and controls. Incubation with hpFSH at a concentration of 30 mIU/mL appeared to increase FSHR expression mainly in patients. Incubation of human spermatozoa with hpFSH overall resulted in an overall deterioration of both progressive and total motility in patients and controls and worse mitochondrial function only in controls. Finally, incubation with FSH increased AKT^473^/tubulin phosphorylation to a greater extent than AKT^308^. FSHR is expressed in the post-acrosomal region, neck, midpiece, and tail of human spermatozoa. Contrary to a previous study, we report a negative effect of FSH on sperm motility and mitochondrial function. FSH also activates the AKT^473^ signaling pathway.

## 1. Introduction

Follicle-stimulating hormone (FSH) is a member of the glycoprotein hormone family, which, in males, is classically known to regulate spermatogenesis. Indeed, it coordinates a series of paracrine events involved in the proliferation and differentiation of germ cells [1]. FSH acts on the Sertoli cells to stimulate self-renewal and the proliferation of spermatogonia by triggering the release of trophic factors. These include activin A, glial cell-derived neurotrophic factor (GDNF), c-kit ligand, bone morphogenetic protein 4 (BMP4), and neuregulins (NRGs) [1]. GDNF−/− mice have no spermatogonia in their testes [2,3]. Therefore, this growth factor is believed to play an important role in regulating self-renewal and maintaining undifferentiated spermatogonia in mammals. However, FSH also induces spermatogonia differentiation. This latter mechanism is believed to occur, on the one hand, from the enhancement of retinoic acid (RA) signaling, thus increasing the expression of Rdh10, Aldh1a1, and Crabp2 gene expression in Sertoli cells and decreasing Crabp1 gene expression in spermatogonia. On the other end, RA induces Sohlh1/2, KL, DMRT, BMP4, and NRGs gene expression in germ cells. All the factors produced by these genes play an important role in spermatogonial differentiation and the induction of meiosis [1].

This recently proposed molecular pathway supports the concept that FSH can support spermatogenesis not exclusively through Sertoli cell-mediated signaling but also through a direct effect on germ cells. This, in turn, should involve the expression of the FSH receptor (FSHR) in these cells.

FSHR is a membrane glycoprotein belonging to the highly conserved family of class A G-protein-coupled receptors (GPCRs). The gene encoding this receptor consists of 10 exons and 9 introns, and it is located on the short arm of chromosome 2 (2p16.3). The protein is made up of 678 amino acids and has been shown to migrate as two separate molecular weight (MW) proteins: one with a low MV (~67 kDa) [4], which comprises a precursor of the mature receptor, and the mature receptor itself, with a high MW (~74 kDa) [5]. Although FSHR was classically believed to be exclusively expressed in gonads and, specifically, in granulosa and Sertoli cells in females and males, respectively, recent data seem to point toward an extra-gonadal expression of FSHR. Indeed, this receptor has been identified in the placenta, umbilical cord vessels, human umbilical vein endothelial cells [6], the endothelial cells of the myometrium [7], adipose tissue, bones [8], and various tumors. These include prostate, thyroid, and ovarian carcinomas, pancreatic neuroendocrine, and pituitary tumors [8].

Evidence supporting FSHR expression in germ cells is scares. Spermatozoa are haploid cells whose main function is to transport and inject the DNA into the oocyte during fertilization. To achieve this, spermatozoa pass through the female genital tract after being deposited in the vagina. This is allowed by their motility and guaranteed by the beating of the flagellum. Mitochondria are the sperm powerhouse. Data indicate that mitochondrial function, as assessed through the mitochondrial membrane potential (MMP), can predict sperm motility [9]. In turn, the higher the MMP, the greater the sperm motility and, therefore, the possibility of fertilization [10].

To the best of our knowledge, only one study has shown that FSHR is expressed in human spermatozoa [11]. The receptor was described in the sperm midpiece with a lower expression level in patients with varicocele compared to healthy men. Incubation with increasing concentrations of FSH enhanced sperm motility in both varicocele patients and controls. The effects were reversed by inhibiting the protein kinase A (PKA) that is activated downstream of the FSHR signaling pathway [11]. These findings suggest that FSH improves sperm motility by acting directly on spermatozoa. If confirmed, this evidence may prompt further physiologic mechanisms by which FSH regulates sperm function as well as a possible role for this hormone in in vitro treatment before assisted reproductive techniques (ART).

On these premises, this study aimed to confirm whether FSHR is expressed in human spermatozoa. Next, we evaluated the effects of increasing concentrations of human purified FSH (hpFSH) on (i) FSHR expression, (ii) sperm motility and mitochondrial function, and (iii) phosphorylation of protein kinase B (PKB or AKT), a mediator of the FSHR signaling pathway [12], at positions 473 (AKT^473^) and 308 (AKT^308^).

As it is currently unknown whether any effects of FSHR, if any, are direct or indirect (i.e., mediated by seminal plasma or the non-germ cell component) at the sperm level, the experiments were conducted in asthenozoospermic patients (patients) and normozoospermic men (controls) using spermatozoa recovered by swim-up, in-toto semen, and semen pellets.

## 2. Results

### 2.1. Is FSHR Expressed in Human Spermatozoa?

Conventional sperm parameters of the four consecutively enrolled subjects used to evaluate FSHR expression are described in Appendix A. In detail, two subjects had normozoospermia, one had oligo-asthenozoospermia, and one had asthenozoospermia.

We found that the FSHR mRNA is expressed in human spermatozoa separated by swim-up (Appendix A). The expression did not show any correlation with any of the conventional sperm parameters. The WB experiments showed a band of approximately 62–67 kDa, corresponding to a protein with a lower MW than that classically expected (~75 kDa). While, on the one hand, this questioned the expression of FSHR in spermatozoa, on the other hand, this may reflect the presence of the low MW FSHR form previously described [4]. Indeed, IF clearly showed its presence as a protein located in the post-acrosomal segment, neck, midpiece, and tail (Figure 1). To further clarify the reliability of the IF results, the experiment was repeated using the same antibody in porcine Sertoli cells and HT29 cells, which were used as positive and negative controls, respectively. As expected, IF showed the presence of IF in Sertoli cells and no signal in HT29 cells, which strengthened the accuracy of the results found by IF in spermatozoa (Appendix A).

### 2.2. Does Incubation with FSH Influence the Expression and Localization of FSHR?

We evaluated whether incubation with hpFSH was able to modulate the expression of FSHR in human spermatozoa from patients and controls. We found an increase in the levels of FSHR protein expression after 60 min of incubation in all patient samples. In controls, the level of FSHR increased in 1 out of the 3 samples tested after incubation. In contrast, the levels of expression remained unchanged in the other two samples (Figure 2). This was partially confirmed by IF (Figure 3A). Indeed, in Patient 4, we did not identify the expression at baseline (Figure 3B, left panel), while it was present after incubation with FSH (Figure 3B, right panel). However, Control 2 and Patient 2 expressed FSHR even before incubation (Figure 3C).

### 2.3. Does Incubation with FSH Influence Progressive and Total Sperm Motility?

Incubation with hpFSH resulted in a concentration-dependent decrease in the percentage of spermatozoa with progressive motility in the in-toto semen. This effect reached statistical significance with a concentration of 30 mIU/mL in asthenozoospermic patients and a concentration of 10 mIU/mL in controls. The percentage of total sperm motility decreased in a concentration-dependent manner in both patients and controls, and the effect reached statistical significance only at a concentration of 30 mIU/mL (Figure 4A).

When the semen pellet was used, all FSH concentrations significantly decreased the percentage of spermatozoa with progressive motility in both patients and controls. The percentage of total motile spermatozoa did not change in controls but significantly decreased in patients with asthenozoospermia at concentrations of 10 and 30 mIU/mL (Figure 4B).

No change was observed with any of the FSH concentrations used in spermatozoa separated by swim-up (Figure 4C).

### 2.4. Does Incubation with FSH Influence Sperm Mitochondrial Function?

The percentage of spermatozoa with H-MMP or L-MMP is shown in Table 1. In in-toto semen, incubation with FSH reduced the percentage of spermatozoa with H-MMP only in normozoospermic men at all concentrations tested. In contrast, no difference in L-MMP or H-MMP was observed in patients with asthenozoospermia or when semen pellets from both patients and controls were used.

### 2.5. Does Incubation with FSH Influence the Phosphorylation of AKT473 and AKT308?

The phosphorylation of AKT473 and AKT308 was measured before and after incubation with hpFSH in patients and controls. Levels were compared with those of tubulin. We found that the p-AKT Ser473/tubulin ratio increased after incubation with hpFSH in most of the patients and all controls. In contrast, the p-AKT Ser308/tubulin ratio increased in a minority of patients and controls, thus suggesting that Ser473 phosphorylation signaling may be targeted by the FSH pathway in human spermatozoa (Figure 5).

## 3. Discussion

FSH is a gonadotropin involved in the proliferation and differentiation of spermatogonia in mammals. A recent review studying the molecular mechanisms by which FSH acts has suggested its direct effects on germ cells. Indeed, FSH downregulates some transcription factors (e.g., Crabp1) in spermatogonia, promoting their differentiation and entry into meiosis to become first-order spermatocytes [1]. The expression of FSHR in human spermatozoa has recently been reported at the level of the sperm midpiece. This study also reported that incubation of swim-up recovered spermatozoa with FSH increases total sperm motility with a concentration of 10 mIU/mL. Since this effect was reversed by a PKA inhibitor, the authors hypothesized that it involves the FSHR [11].

Since this topic has never been investigated before, the present study was undertaken to confirm the expression of FSHR in human spermatozoa. We found that both FSHR mRNA and protein are expressed in spermatozoa recovered by swim-up as well as in the sperm neck, midpiece, and tail. Additionally, WB showed a protein with a lower MW than that at which FSHR is found classically (~75 kDa). This suggests that the FSHR expressed in spermatozoa selected by swim-up is different from that found in Sertoli cells. This may result from post-translational modifications of the FSHR in spermatozoa, post-translation cleavage, relative charges, or the presence of a differentially glycosylated form of the FSHR in spermatozoa [5]. Alternatively, it may result from a different glycosylation process that expresses a differently glycosylated form of FSHR with a lower MW (~52–69 kDa) [5]. Accordingly, a low-MW FSHR protein has been described [4].

Subsequently, we evaluated the effects of increasing concentrations of hpFSH on sperm motility and mitochondrial function by evaluating the percentage of spermatozoa with H-MMP or L-MMP. In an attempt to better understand the effects of FSH, the experiments were carried out in patients with asthenozoospermia and normozoospermic men using in-toto semen, semen pellets, and spermatozoa recovered by swim-up. We found that FSH worsened sperm motility, which was more evident in the sperm pellet. Indeed, while only the concentration of 30 mIU/mL showed a negative effect in the in-toto semen of asthenozoospermic patients, all concentrations tested were able to significantly reduce progressive motility in both patients and controls. The reason for the different results using in-toto semen and semen pellets is not known. We hypothesize that it could relate to the presence of an “FSH-like factor” whose presence was reported in the human seminal plasma many years ago [13]. This protein seems to have inhibitory properties, thus acting as a regulator of FSH local effects [14].

hpFSH had no significant effect on spermatozoa recovered by swim-up except on progressive motility, which decreased. The interpretation of these findings is not immediately clear. Some hypotheses may be proposed. First, the swim-up technique allows for recovering the fraction of spermatozoa with the best motility. Due to their better quality, this population of spermatozoa may resist the lowering effects that FSH seems to have on motility. Furthermore, the role of the capacitating medium that was used for swim-up, which per se enhances sperm motility, cannot be excluded. Moreover, swim-up-recovered spermatozoa lack seminal fluid, which may interfere with the effect of FSH. Finally, the pellet differs from the swim-up also for the presence of somatic cells such as leukocytes and epithelial cells. Some leukocyte subpopulations express the FSHR. In particular, this receptor has been identified in monocytes, where incubation with FSH stimulates tumor necrosis factor α (TNFα) release [15]. The negative impact that TNFα has on sperm motility, vitality, and DNA fragmentation has already been reported [16]. This may explain the reduction of progressive motility in the pellet samples incubated with FSH.

Lastly, FSH mediates its effects in spermatozoa by prominently triggering AKT473 signaling. In Sertoli cells, FSH is known to induce proliferative and anti-apoptotic signals through the AKT pathway [17]. However, another direct effect of FSH involves calcium channels. In Sertoli cells, by triggering the protein kinases A and C, FSH induces intracellular calcium release and influx from T-type calcium channels [17]. Intracellular calcium oscillations impact sperm motility, though they mediate a stimulatory effect [18].

It is noteworthy to mention that several single nucleotide polymorphisms (SNPs) of the FSHR gene have been recognized as capable of influencing receptor function. The HapMap database identified about 900 different SNPs for the FSHR gene (http://hapmap.ncbi.nlm.nih.gov, accessed on 1 March 2022). Some of them, such as the FSHR c. 2039 A/G (p. Asp680Ser) (rs 6166), are known to influence the efficiency of signal transduction, and others, such as the FSHR-29 G/A, may impact gene expression [19]. These SNPs may hypothetically influence the effects of FSH on spermatozoa in vitro. This may add further evidence for the discrepancy between the present data and those published previously [11]. However, the stark contrast between our results and those of Panza and colleagues [11] highlights the need to further explore this topic.

The expression of FSHR in spermatozoa allows speculating on new possible physiologic mechanisms by which FSH can regulate spermatogenesis and suggests a possible direct effect of this hormone on germ cells. Accordingly, FSH has been reported in human seminal plasma at a concentration variable between 4.4 and 35.4 mIU/mL [20,21] which supports the possibility of a direct effect of FSH on spermatozoa. The clarification of this issue is relevant since FSH is used for the treatment of male infertility. A pioneering study on hypophysectomized rats has shown, after treatment with hpFSH and/or testosterone, that while FSH sustains the proliferative phase of spermatogenesis, testosterone plays a role in the differentiation phase from spermatid to spermatozoon [22]. However, FSH is administered to oligozoospermic patients within normal FSH serum levels (in Italy, the cut-off for FSH prescription is <8 mIU/mL), and meta-analytic studies indicate dose-dependent effectiveness not only on sperm concentration and total sperm count but also on progressive motility [23]. FSH treatment seems to improve sperm DNA fragmentation as well [24]. Noteworthy, FSHR−/− mice show a significant increase in the percentage of spermatozoa with fragmented DNA compared to wild-type animals [25], which further highlights the protective role of FSH on sperm DNA integrity. Interestingly, this mouse model also shows altered sperm motility compared to the wild-type [26], indicating a role for the FSH/FSHR pathway in sperm motility.

The use of FSH for the in vitro treatment of sperm aliquots to be used for ART has been suggested [11]. However, our data do not seem to support this indication. Further studies are definitively needed to better understand this aspect.

In conclusion, we found that FSHR is expressed in the post-acrosomal region of the sperm head, neck, midpiece, and tail. FSH shows deteriorating effects on human sperm motility and mitochondrial function, and it induces AKT473 phosphorylation in human spermatozoa. Since the presence of FSH in human seminal plasma has already been reported [20,21], our findings support a potential direct effect of FSH on differentiated spermatozoa. We also speculate that in vitro treatment with FSH should not be suggested for ART.

## 4. Materials and Methods

### 4.1. Experimental Design

The first part of the experimental design aimed to confirm whether FSHR is expressed in human spermatozoa. This was accomplished by evaluating mRNA and protein expression in four consecutively enrolled subjects. The only criterion for their enrollment was the presence of spermatozoa in their semen. Protein expression was evaluated both by Western blot (WB) and immunofluorescence (IF). Sertoli cells and the HT29 cell line (isolated from a primary colorectal adenocarcinoma in 1964) were used as positive and negative controls, respectively.

Next, we evaluated the effects of hpFSH (0 and 30 mIU/mL) in swim-up-separated human spermatozoa on FSHR protein expression and localization by WB and IF, respectively. Subsequently, we investigated whether incubation with hpFSH at concentrations of 0, 1, 10, and 30 mIU/mL had any effect on progressive and total sperm motility and mitochondrial function. For this experiment, we used swim-up-separated spermatozoa. Furthermore, to evaluate the role of seminal plasma and the non-germinal component of the semen fluid in the observed findings, we repeated the latter experiments using in-toto semen and the semen pellet.

Finally, WB analyzed the phosphorylation of AKT473 and AKT308 after incubation with hpFSH (0 and 30 mIU/mL) on swim-up separated spermatozoa.

The semen samples were not pooled together, as each donor sample was analyzed separately. Furthermore, all the experiments were performed in patients with asthenozoospermia and normozoospermic controls. A complete flowchart of the experimental design is shown in Figure 6.

### 4.2. Patients Selection

This study was conducted on men who were referred to the Division of Endocrinology, Metabolic Diseases, and Nutrition, University of Catania, for semen analysis. We recruited 24 Caucasian men between the ages of 18 and 43 with normal sperm count and morphology and no leukocytospermia. Based on the sperm motility, participants were divided into two groups: men whose progressive sperm motility was greater than 32%, who served as controls (n = 11), and patients with progressive sperm motility lower than 32%, who made up the asthenozoospermic group (n = 13) (WHO 2010). Samples of patients with azoospermia, hyper-viscosity, male accessory gland infection/inflammation, genetic syndromes, previous or current exposure to chemo- and/or radiotherapy, retrograde ejaculation, hypogonadism, hormone therapy (e.g., on FSH, selective estrogen receptor modulators, testosterone replacement therapy, aromatase inhibitors, etc.), or antioxidants were excluded.

The conventional sperm parameters of patients and controls are shown in Table 2.

### 4.3. Semen Collection and Processing

Semen samples were collected by masturbation and placed in a sterile container. The collection took place in a private room near the laboratory of seminology of the Division of Endocrinology, Metabolic Diseases, and Nutrition, University of Catania (Catania, Italy). Each patient recruited for the study was asked to have sexual abstinence between three and five days before sample collection. The semen samples were then kept at room temperature (20–22 °C) and analyzed immediately after their complete liquefaction according to the criteria suggested by the WHO (WHO 2010).

After liquefaction, a 1 mL aliquot of each sample was subjected to the swim-up separation protocol. Then, it was washed using K-SIMS-50 sperm medium (Sydney IVF, William A. Cook, Queensland, Australia), which is rich in glucose and centrifuged for 15 min at 500× *g*. After removing the supernatant, 1 mL of K-SIMS-50 sperm medium was placed over each aliquot of the semen pellet. The tubes were incubated at 37 °C in a 5% CO_2_ atmosphere for 45 min, tilted at an angle of approximately 45°. At the end of the incubation, 1 mL of supernatant was gently removed from each tube [27]. This upper fraction contains highly motile spermatozoa. Indeed, glucose-enriched medium prompts the best spermatozoa to migrate upward.

Spermatozoa were then washed in 1× phosphate-buffered saline (PBS) and used for RNA extraction, protein evaluation, or incubation experiments.

Since the protocol involved evaluating the effects of hpFSH incubation in the in-toto semen or semen pellet, an aliquot of the semen from patients and controls was incubated directly with hpFSH (in-toto semen) or after centrifugation (5 min at 500× *g*) (semen pellet).

### 4.4. Treatment

The three samples thus obtained (swim-up, in-toto semen, and semen pellet) were then subsequently used for incubation. For each concentration used in this study (FSH 0, 1, 10, and 30 mIU/mL), an aliquot of 10 × 10^6^ spermatozoa was washed with a non-supplemented, non-capacitating Earle’s balanced salt solution and incubated at 37 °C in a 5% CO_2_ atmosphere for 60 min with hpFSH (Fostimon, IBSA Farmaceutici Italia Srl, Lugano, Switzerland). We then analyzed FSHR expression, progressive and total sperm motility, mitochondrial function, and phosphorylation of AKT^473^ and AKT^308^.

### 4.5. Motility Assessment

In order to evaluate sperm motility, immediately after liquefaction of the semen, 10 µL of an undiluted, well-mixed semen sample was loaded into the center of a clean Makler counting chamber, maintained at a temperature of 37 °C, gently covered with a coverslip, and examined using 200× magnification. Sperm motility was assessed in 200 random spermatozoa and evaluated as progressive or non-progressive motility.

According to the WHO Manual, progressive motility is defined by the percentage of spermatozoa that are actively moving, either linearly or in a large circle, regardless of speed. Non-progressive motility is defined as the percentage of spermatozoa that move in all other patterns of motility without progressing in the chamber (e.g., rotating in small circles, flagellar force barely capable of moving the sperm head, presence of a slight flagellar beat, etc.). Finally, immotile spermatozoa are those that do not show any movement. The endpoints of the study were progressive and total motility; the latter was calculated as the sum of progressive and non-progressive motility. Motility was evaluated by the same well-trained seminologist (N.B.) with more than 20 years of experience in semen analysis at the seminology laboratory of the Division of Endocrinology, Metabolic, and Nutritional Diseases, University of Catania.

### 4.6. RNA Extraction, Reverse Transcription, and Quantitative Real-Time PCR

RNA extraction and real-time PCR (RT-PCR) were used to assess whether FSHR mRNA is present in human spermatozoa. Total RNA was extracted with Trizol^®^ Reagent (Life Technologies, Monza, Italy) according to the manufacturer’s instructions. The concentration and purity of RNA were assessed by the Eppendorf Biophotometer. cDNA reverse transcription was performed for each sample using a cDNA synthesis kit (Thermo Scientific Maxima First Strand cDNA Synthesis Kit for RT-qPCR), according to the manufacturer’s instructions. Briefly, total RNA was extracted from the samples using the Trizol reagent (Sigma-Aldrich, Milan, Italy) and quantified by reading the optical density at 260 nm. In particular, 2.5 µg of total RNA was reverse transcribed (RT) (Thermo Scientific, Waltham, MA, USA) to a final volume of 20 μL. The qPCR was performed using 50 ng of the cDNA prepared by RT and an SYBR Green Master Mix (Stratagene, Amsterdam, The Netherlands—Agilent Technology). This was done in an AriaMx Version 1.71 (Stratagene, Amsterdam, The Netherlands—Agilent Technology), using FAM for detection and ROX as the reference dye. The mRNA level of each sample was normalized by β-actin mRNA and expressed as fold changes. The following primers were used: forward sequence (5′–3′): FSHR TGAGTATAGCAGCCACAGATGACC; reverse sequence (5′–3′): TTTCACAGTCGCCCTCTTTCCC; forward sequence (5′–3′): actin ATGGTGGGTATGGGTCAGAA; reverse sequence (5′–3′): CTTCTCCATGTCGTCCCAGT. The thermal cycling conditions were 1 cycle at 95 °C for 5 min, followed by 45 cycles at 95 °C for 20 s and 58 °C for 30 s. The data necessary to perform a comparative analysis of gene expression were obtained using the 2^−ΔΔCT^ method.

### 4.7. Western Blot Analysis

Total cell lysates were collected in RIPA lysis buffer (Santa Cruz Biotechnology Inc., Santa Cruz, TX, USA). The mixture was centrifuged at 1000× *g* (Eppendorf, NY, USA) for 10 min, the supernatant was collected, and the total protein content was measured by the Bradford method [28]. Aliquots of samples were stored at −20 °C to perform a small-scale affinity purification of FSHR by immunoprecipitation (IP) analysis, as previously described [29]. Briefly, we transferred 500 μg of total cell protein to a 1.5 mL microcentrifuge tube, added 5 μg of mouse anti-FSHR primary antibody (NBP2-36489, clone 6E8.2F5, Novusbio, CO, USA), and incubated overnight at 4 °C on a rocking platform. Then, we added 25 μL of Protein A/G PLUS-Agarose (Santa Cruz Biotechnology Inc., Dallas, TX, USA) and incubated at 4 °C on a rocking platform for 3 h. We collected immunoprecipitates by centrifugation at 1000× *g* for 5 min at 4 °C. The supernatants were carefully aspirated and discarded, and the pellets were washed four times with a 1.0 mL RIPA Lysis Buffer System (Santa Cruz Biotechnology Inc., Dallas, TX, USA), repeating the centrifugation step each time. After the final wash, we aspirated and discarded the supernatants and resuspended the pellets in 20 μL of sample electrophoresis buffer containing 50% glycerol, 20% sodium dodecyl sulfate (SDS), 0.5 M Tris–HCl (pH 6.8), 5% 2-mercaptoethanol, and 0.02% bromophenol blue. The buffer was boiled for 5 min, and the pellets were loaded onto a 4–12% (*w/v*) SDS-PAGE gel [30]. The proteins were separated and transferred to nitrocellulose membranes using an iBlotTM 2 Dry Blotting System (Thermo Fisher, Waltham, MA, USA) [29]. After blocking the membrane with 5% dry milk powder in 10 mM Tris-HCl (pH 8), 0.5 M NaCl, and 1% Tween-20 (TBS), the membranes were incubated with a mouse anti-FSHR primary antibody (NBP2-36489, clone 6E8.2F5, Novusbio, CO, USA, dilution factor 1:300). After being washed with TBS containing 1% Tween-20, the blots were incubated with anti-mouse peroxidase-conjugated secondary antibodies (HRP) (1:5000, Santa Cruz Biotechnology Inc., Dallas, TX, USA) and developed using enhanced chemiluminescence (ECL; Bio-Rad, Hercules, CA, USA), according to the manufacturer’s instructions. Positive controls were porcine Sertoli cells. At the same time, to study the AKT pathway, aliquots of 70 μg protein were prepared from the total cell extract, then run and blotted, as previously reported [12], with rabbit 13038 anti-phospho-AKT (Thr308) (dilution factor 1:1000) (Cell Signaling Technology, Inc., Danvers, MA, USA), and rabbit 9271 anti-phospho-AKT (Ser473) (dilution factor 1:1000) (Cell Signaling Technology, Inc., Danvers, MA, USA). Primary antibody binding was then detected by incubating membranes with horseradish peroxidase conjugated anti-rabbit secondary antibodies (Sigma-Aldrich Co., St. Louis, MO, USA, 1:5000). ECL analysis was performed, as previously reported.

### 4.8. Immunofluorescence Analysis

The spermatozoa were spread onto microscope slides, dried in air at room temperature (RT), and fixed in absolute methanol for 10 min at −20 °C. Fixed cells were blocked with 0.5% BSA in PBS (Sigma-Aldrich, St. Louis, MO, USA) for 1 h before exposure to the mouse anti-FSHR primary antibody (NBP2-36489, clone 6E8.2F5, Novusbio, CO, USA, dilution factor 1:200) at 4 °C, overnight. Subsequently, the cells were incubated with a donkey anti-mouse IgG (H&L), a DyLight^®^ 488 conjugated secondary antibody (1:500, Thermo Fisher Scientific, Waltham, MA, USA), treated with RNAse (10 mg/mL, Sigma-Aldrich, St. Louis, MO, USA), and counterstained for 1 min with 4′,6-diamidino-2-phenylindole (DAPI, Sigma-Aldrich, St. Louis, MO, USA). Cells were mounted with ProLong^®^ Gold antifade reagent (Molecular Probes, NY, USA). FSHR-positive cells were visualized under the BX-41 microscope (Olympus, Tokyo, Japan) equipped with a fluorescence camera (F-Viewer, Olympus, Tokyo, Japan). Images were processed with Cell F imaging software (Olympus, Tokyo, Japan). The same analysis was repeated in prepubertal porcine Sertoli cells [31] (Mancuso et al., 2022) and HT29 cell lines (Istituto Zooprofilattico Ubertini, Brescia, Italy), which served as positive and negative controls for FSHR expression, respectively.

### 4.9. Evaluation of Sperm Mitochondrial Membrane Potential

MMP was evaluated using a lipophilic probe 5,5′,6,6′-tetrachloro 1,1′,3,3′tetraethylbenzimidazolylcarbocyanine iodide (JC-1, DBA s.r.l, Milan, Italy), which can selectively penetrate mitochondria. Briefly, an aliquot containing 1 × 10^6^/mL of spermatozoa was incubated with JC-1 in the dark for 10 min at 37 °C. At the end of the incubation period, the cells were washed in PBS and analyzed. JC-1 exists in monomeric form, emitting at 527 nm, but it can form aggregates emitting at 590 nm. Therefore, the fluorescence reversibly changes from green to orange as the mitochondrial membrane becomes more polarized. In viable cells with normal membrane potential, JC-1 is found in the mitochondrial membrane in the form of aggregates that emit an orange fluorescence, while in cells with low membrane potential, it remains in the cytoplasm in monomeric form and has a green fluorescence.

### 4.10. Power Analysis

Sperm motility was chosen for power calculation as it was the primary outcome. According to the 2010 WHO manual, a progressive motility of 32% has been hypothesized. An absolute decrease of 0% was considered of sufficient importance to modify clinical practice. The study required 6 participants (3 per group) to maintain 90% power and detect an absolute difference of 5%.

### 4.11. Statistical Analysis

The results are reported as means ± SEM throughout the study. For functional analysis, we reported the percentage of change from FSH 0 mIU/mL. The normal distribution of the variables was evaluated with the Shapiro–Wilk test. Differences between groups in progressive and total sperm motility for spermatozoa with low (L-MMP) or high (H-MMP) MMP were analyzed by one-way analysis of variance (ANOVA) and the Tukey-Kramer post-hoc test. Statistical analysis was performed using MedCalc Software Ltd. (Ostend, Belgium; https://www.medcalc.org; accessed on 17 June 2021, Version 19.6–64 bit). A *p*-value lower than 0.05 was accepted as statistically significant.

### 4.12. Ethical Approval

This study was conducted at the Division of Endocrinology, Metabolic Diseases, and Nutrition of the University-Teaching Hospital Policlinico “G. Rodolico—San Marco”, University of Catania (Catania, Italy). The protocol was approved by the internal Institutional Review Board. A written informed consent was obtained from each participant after a full explanation of the purpose and nature of all procedures used. The study was carried out according to the principles expressed in the Declaration of Helsinki.

## Figures and Tables

**Figure 1 ijms-24-06536-f001:**
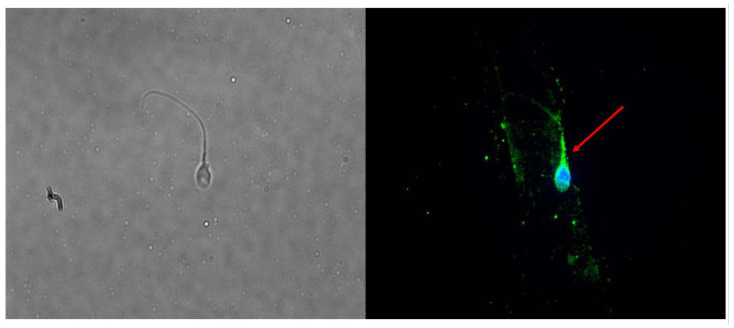
Immunolocalization of the follicle-stimulating hormone receptor (FSHR) protein in human spermatozoa. Immunofluorescence (IF) analysis of spermatozoa. The protein is present in the post-acrosomal segment, neck, midpiece, and tail of the spermatozoon. Magnification 100×. IF is indicated by the red arrow.

**Figure 2 ijms-24-06536-f002:**
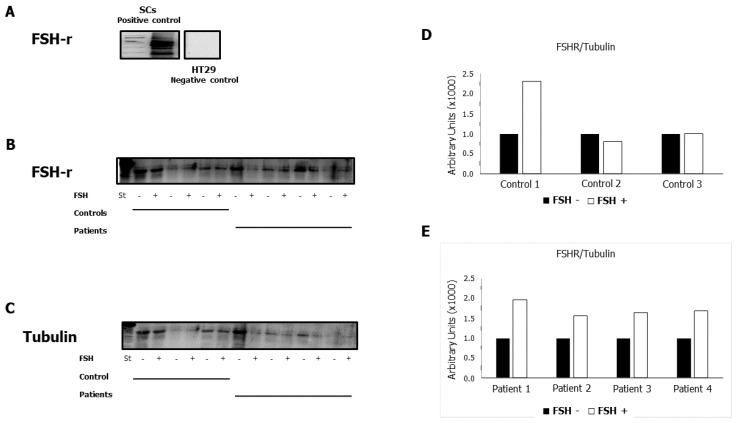
Expression of follicle-stimulating hormone receptor (FSHR) protein in human spermatozoa from patients and controls before and after incubation with hpFSH. Western blot (WB) analysis of FSHR in positive [Sertoli cells (SCs)] and negative (HT29) controls (**A**). WB analysis of FSHR in spermatozoa from patients and controls (**B**). WB analysis of tubulin in spermatozoa from patients and controls (**C**). Densitometric analysis of the FSHR/tubulin ratio in spermatozoa from controls (**D**). Densitometric analysis of the FSHR/tubulin ratio in spermatozoa from patients (**E**). The data represent the mean ± standard deviation of three independent experiments, each performed in triplicate.

**Figure 3 ijms-24-06536-f003:**
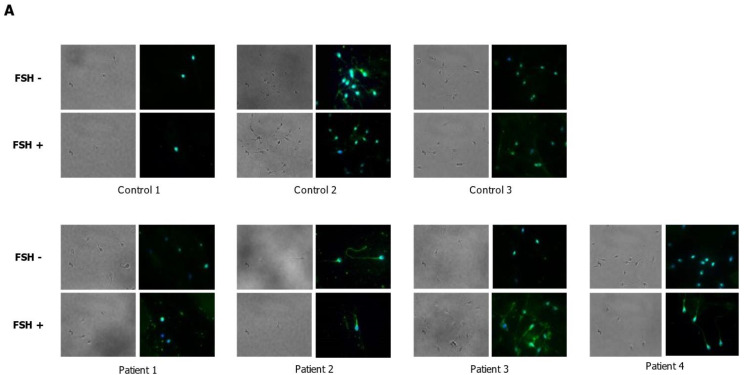
Immunolocalization of the follicle-stimulating hormone receptor (FSHR) protein in human spermatozoa after incubation with hpFSH. Immunofluorescence (IF) analysis of the FSHR in spermatozoa of controls and patients (**A**). Magnification from Patient 4: before incubation, the protein was not identified (Left panel). After incubation, it appeared in the post-acrosomal region, neck, midpiece, and tail (Right panel) (**B**). Magnification from Control 2 (Left panel) and Patient 2 (Right panel): the protein appears expressed even before incubation with FSH (**C**). Magnification 100×. IF is indicated by the red arrows in Panels B and C.

**Figure 4 ijms-24-06536-f004:**
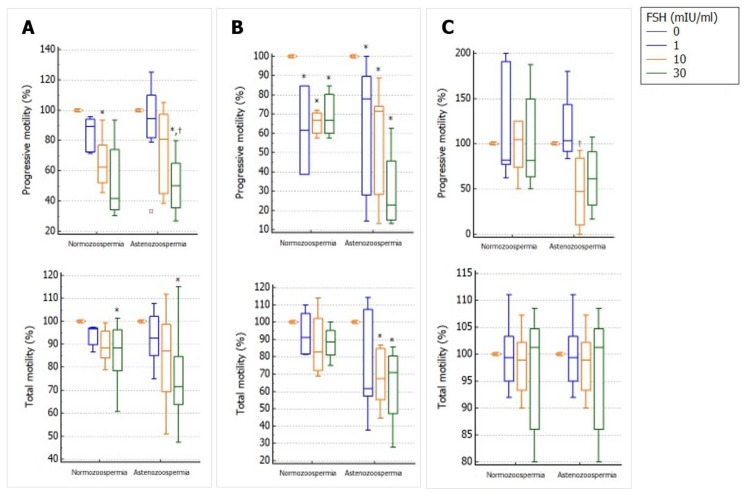
Effects of increasing concentrations of follicle-stimulating hormone (FSH) on *in-toto* semen sample (**A**), semen pellet (**B**), and spermatozoa recovered by swim-up (**C**) from normozoospermic men and asthenozoospermic patients. Data are reported as the percentage change from the control (FSH 0 mIU/mL), using boxplots. The line inside the box indicates the median, the end of the box the interquartile range, and the values outside the box show the minimum and maximum values. The different colors indicate the different doses of FSH used for sample incubation. * *p* < 0.05 vs. 0 mIU/mL; ^†^ *p* < 0.05 vs. 1 mIU/mL.

**Figure 5 ijms-24-06536-f005:**
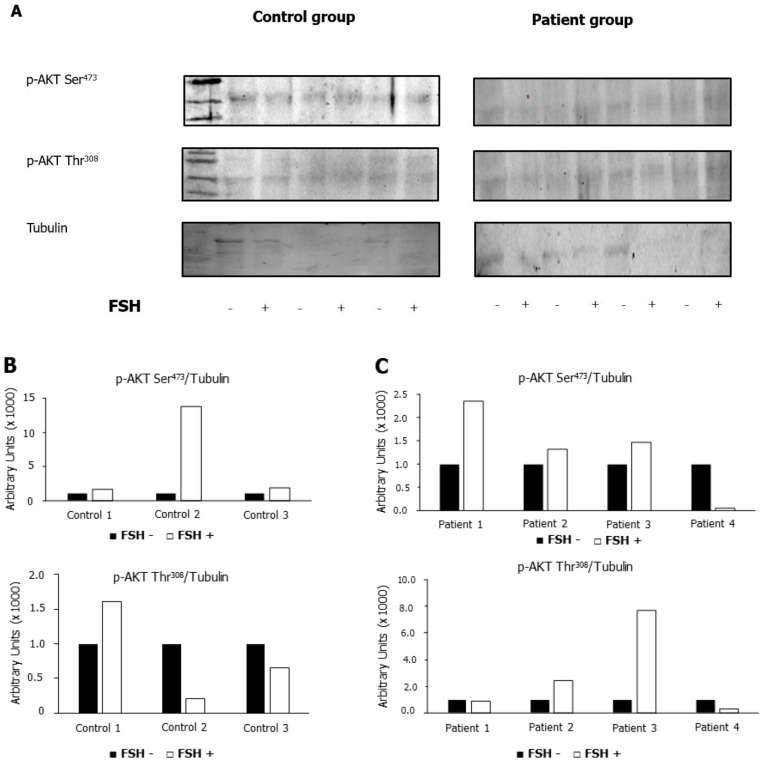
Phosphorylation of protein kinase B (AKT) in Serine 473 and Threonine 308. Immunoblots (**A**) and densitometric analysis (**B**,**C**) of p-AKT Ser^473^/tubulin ratio and p-AKT Thr^308^/tubulin ratio in patients and controls. p-AKT Ser^473^/Tubulin increased after incubation with hpFSH in most of the patients and controls. p-AKT Ser^308^/tubulin ratio increased in a minority of patients and controls.

**Figure 6 ijms-24-06536-f006:**
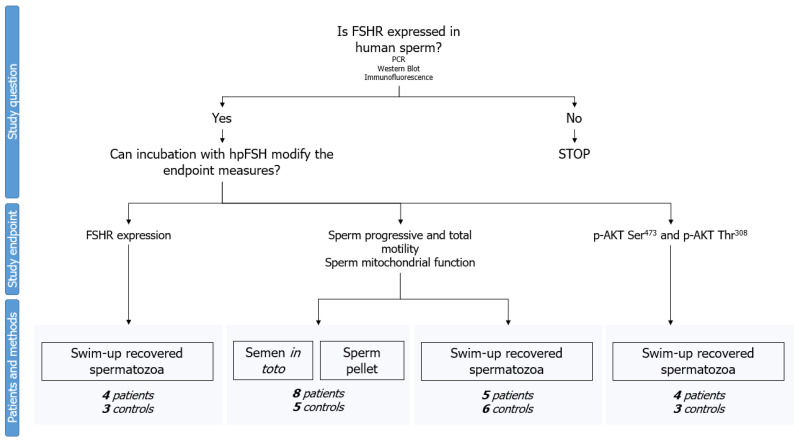
Flow chart of the experimental design. After confirmation of follicle-stimulating hormone receptor (FSHR) expression in human spermatozoa by polymerase chain reaction (PCR), Western blot, and immunofluorescence, experiments were conducted in men with normozoospermia and patients with asthenozoospermia using spermatozoa separated by swim-up, *in-toto* semen, or semen pellets.

**Table 1 ijms-24-06536-t001:** Percentage (mean ± SEM) of spermatozoa with high or low mitochondrial membrane potential reported as percentage change from the control (FSH 0 mIU/mL).

	*In-Toto* Semen		Semen Pellet		Spermatozoa Separated by Swim-Up	
FSH Concentration (mIU/mL)	Asthenozoospemia	Normozoospermia	Asthenozoospemia	Normozoospermia	Asthenozoospemia	Normozoospermia
*High mitochondrial membrane potential (%)*						
0	100	100	100	100	100	100
1	103.7 ± 2.4	94.2 ± 3.3 *	104.6 ± 3.9	92.9 ± 9.7	100.2 ± 0.9	100.5 ± 0.3
10	105.8 ± 6.8	92.9 ± 2.3 *	94.1 ± 5.0	102.4 ± 1.1	99.4 ± 0.3	97.9 ± 1.5
30	92.6 ± 3.6	92.0 ± 3.8 *	87.8 ± 5.0	83.4 ± 19.3	99.4 ± 0.1	101.2 ± 1.6
*Low mitochondrial membrane potential (%)*						
0	100	100	100	100	100	100
1	89.3 ± 7.5	114.7 ± 19.1	88.8 ± 11.5	118.2 ± 26.7	100.7 ± 18.6	90.8 ± 7.1
10	99.6 ± 23.1	179.9 ± 60.8	110.4 ± 18.7	73.2 ± 10.0	120.6 ± 0.8	124.5 ± 30.6
30	119.7 ± 8.2	188.3 ± 53.8	181.0 ± 38.3	141.7 ± 71.6	123.9 ± 10.1	91.9 ± 26.7

***Legend:*** FSH, follicle-stimulating hormone; * *p* < 0.05 vs. FSH 0 mIU/mL.

**Table 2 ijms-24-06536-t002:** Conventional sperm parameters of asthenozoospermic patients (patients) and normozoospermic men (controls) enrolled in this study.

Parameters	All Cohort		Samples Used for *In-Toto* Semen and Pellet		Samples Used for Swim-Up	
	Patients (n = 13)	Controls (n = 11)	Patients (n = 8)	Controls (n = 5)	Patients (n = 5)	Controls (n = 6)
Sperm concentration (million/mL)	41.5 ± 38.7	59.0 ± 29.4	39.8 ± 32.4	68.0 ± 25.1	44.2 ± 51.3	50.0 ± 33.4
Total sperm count (million/ejaculate)	108.8 ± 95.0	169.8 ± 109.8	118.4 ± 113.2	224.5 ± 104.9	93.3 ± 64.3	115.0 ± 92.9
Progressive sperm motility (%)	21.9 ± 7.8 *	33.9 ± 5.6	26.4 ± 6.2	33.0 ± 6.7	14.8 ± 3.3 *	34.8 ± 4.8
Total sperm motility (%)	59.0 ± 12.7 *	69.2 ± 5.9	65.3 ± 11.7	73.0 ± 5.7	49.0 ± 6.5 *	65.4 ± 3.2
Spermatozoa with normal morphology (%)	7.2 ± 2.4 *	10.8 ± 3.7	8.3 ± 2.3	11.4 ± 5.2	5.4 ± 1.1 *	10.2 ± 1.8
Leukocyte concentration (million/mL)	0.6 ± 0.5	0.8 ± 0.5	0.4 ± 0.3	0.7 ± 0.4	0.8 ± 0.6	0.9 ± 0.7

Results are expressed as mean ± standard deviation. * *p* < 0.05 vs. the respective controls by student *t*-test.

## Data Availability

Data are available upon request to the corresponding author.

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
