# Peer review of "Effects of Follicle-Stimulating Hormone on Human Sperm Motility In Vitro"

_ijms, 2023, doi:10.3390/ijms24076536_

Round 1
Reviewer 1 Report
General Comment
The paper by Cannarella et al., entitled “Effects of follicle-stimulating hormone on human sperm motil-2 ity in-vitro” aims to know whether the follicle-stimulating hormone (FSH) receptor 11 (FSHR) is expressed in human spermatozoa and the effects of FSH incubation on sperm function.
A great part of this objective, however, has been studied before in the paper by Panza et al., 2020. Moreover, results obtained in the present paper about sperm motility are contrary to those obtained by the previous work by Panza et al., 2020, without providing any convincing and data-based scientific explanations.
Therefore, this raises an argument against the present manuscript
The present paper lacks of scientific interest and originality, and the overall merit is very low.
Specific comments:
-Authors do not specified the method they used to evaluate sperm motility, being this sperm function essential in the present work as it is included in the title.
-Table 1: The standard deviations are quite big (for instance, 41.5 ± 38.7). Some explanation? It seems quite difficult to get any warranted conclusion from these data.
Supplementary Figure 1. Legend of this Figure should be rewritten to clarify what can be seen in the graphs or images. There are two types of graphs in this Figure and it is not specified which one is used in each part. The B part is difficult to understand, as there is not clue what is the left or the right images, the lanes are not labeled and some molecular markers below 75 kDa are missed, so the reader can see what the authors mention in their manuscript about the MW of the band. Also, in the figure legend is missed the information about what type of samples are used, control? Patients?
- Figure 2 only shows 1 spermatozoon. It is difficult to guess a clear pattern of immunofluorescence with only 1 cell.
- Authors state (lanes 306-307) that “an increase in the 306 levels of FSHR protein expression after 60 minutes of incubation in all patient samples”. However, this is not visualized at all in the Figure 3B. This referee cannot see any increase in any patient. In addition, the figures of western blot are so small and not clearly labeled their lanes that it cannot be properly visualized.
-Therefore, the histograms included in figure 3 seem not to be related to the western blot images. For instance, in D, the control 1 has more than 2-fold increase after FSH treatment; this is not seen in the related western blot in B, at least this referee visualizes not change o, if any, a decrease after FSH. This should be well explained.
- How can it be possible that expression levels of a protein can increase in spermatozoa? and even more challenging, after only 60 minutes?. This assumption is difficult to scientifically be understood (lanes 306-308).
-In legend of Figure 4 it is stated that “After incubation, it appeared in the post-acrosomal region, neck, mid-329 piece, and tail”, but the images this can only be seen in 1 patient, or maybe 2...
-The type of graphs in Figure 5 are not explained in the legend.
- In lanes 355- 356 is mentioned “In in-355 toto semen, incubation with FSH reduced the percentage of spermatozoa with H-MMP 356 only in normozoospermic men at all concentrations tested”. However, in Table 2, this reduction although significant is so small. This is important because in other data of the table, there are bigger effects (semen pellet asthenozoospermia there is a reduction from 100 to 87,8 and is not mentioned.
Besides, the seemed reduction in H-MMP mentioned by the authors in toto semen from normozoospermic men is not related with a significant increase in the low-MMP.
- Results about AKT phosphorylation are unclear and difficult to convince with the western blot presented. It is unwarranted any conclusion based on these images. It is really hard to visualize AKT band and even harder to see any effect.
-Authors are not convincing when explain the different molecular weight of the band obtained in their western blot. Panza et al (2020) previously showed that the MW of the FSH-R is found about 75 kDa.
-In order to explain their results authors used many hypothesis (Discussion), but convincing data are not provided nor newest data, apart from the already published by Panza et al., 2020.
Author Response
Answers to the Reviewer #1 comments
Manuscript ID ijms-2223251
Comment 1: The paper by Cannarella et al., entitled “Effects of follicle-stimulating hormone on human sperm motil-2 ity in-vitro” aims to know whether the follicle-stimulating hormone (FSH) receptor 11 (FSHR) is expressed in human spermatozoa and the effects of FSH incubation on sperm function. A great part of this objective, however, has been studied before in the paper by Panza et al., 2020. Moreover, results obtained in the present paper about sperm motility are contrary to those obtained by the previous work by Panza et al., 2020, without providing any convincing and data-based scientific explanations. Therefore, this raises an argument against the present manuscript. The present paper lacks of scientific interest and originality, and the overall merit is very low.
Answer to comment 1: We thank this Reviewer for the time spent reading our article. We cannot agree with his/her opinion for the following reasons:
- A great body of literature has recently focused on the extra-Sertolian (or extra-granulosa cell) expression of the FSHR. Actually, this receptor has been identified in the placenta, umbilical cord vessels, human umbilical vein endothelial cells, myometrial endothelial cells, adipose tissue, bones, and various tumors. These include prostate, thyroid, and ovarian carcinomas, pancreatic neuroendocrine, and pituitary tumors (lines 48-54). FSHR showed functional activity in these tissues. As an example, FSHR is expressed in the osteoclasts, where FSH triggers osteoclast metabolism and bone turnover, offering a new mechanism of post-menopausal bone remodeling beside hypoestrogenism, in turn providing new insights for alternative therapeutic approaches of osteoporosis (doi: 10.1016/j.bbrc.2012.04.104; doi: 10.1016/j.bbrc.2010.02.112; doi: 10.1016/j.bbrc.2007.07.081. Furthermore, the FSHR is expressed in adipocytes, and it triggers their metabolism (doi: 10.1194/jlr.M025403). High FSH levels have been suggested to contribute to obesity in post-menopausal women. Taking all this into account, this field of research seems very exciting and prompts new fascinating mechanisms and therapeutic approaches.
- Concerning the expression of FSHR in human spermatozoa, this also represents a new and interesting field that needs to be fully explored. Indeed, proving FSHR expression and elucidating the reasons for its presence in these cells, can ultimately result in a better understanding of the effects of FSH treatment in infertile patients in vivo. Finally, this research field can also clarify the possible role of FSH use in vitro to prepare spermatozoa to be used for assisted reproductive techniques.
- The study by Panza and colleagues is the only one that, so far, has reported the expression of FSHR in spermatozoa. Since this is a novel finding, its confirmation is a very important step. Furthermore, the study by Panza and colleagues has a different design than ours, since these authors investigated patients with varicocele. In contrast, we focused on patients with asthenozoospermia. In addition, the study by Panza and colleagues has several drawbacks. The main weakness is that they only measured total sperm motility and not progressive motility. Total motility is known to be of limited value in clinical practice. Moreover, they pooled samples from different patients to perform their experiments and this is a major drawback. In contrast, the experimental design we have chosen closely follows the clinical pathway. Therefore, our data have a higher adherence to clinical practice.
Comment 2. Authors do not specified the method they used to evaluate sperm motility, being this sperm function essential in the present work as it is included in the title.
Answer to comment 2: We appreciated this suggestion. We reported that the WHO 5th manual was used for semen analysis. Based on your comment, we added to section 2.5, which explains how sperm motility was evaluated. A well-trained and experienced Seminologist (author N.B.) – with more than 20 years of experience in semen analysis, evaluated sperm motility (please see lines 149-161).
Comment 3: Table 1: The standard deviations are quite big (for instance, 41.5 ± 38.7). Some explanation? It seems quite difficult to get any warranted conclusion from these data.
Answer to comment 3: These standard deviations are common to the vast majority of articles reporting conventional sperm parameters (for example, see Table 1 of the article doi: 10.1111/andr.13012). The reason is given by the great variability of these parameters that is both intra- and inter-individual (Fertil Steril. 1985;44:396-400; please also refer to the WHO manual for semen analysis).
Comment 4: Supplementary Figure 1. Legend of this Figure should be rewritten to clarify what can be seen in the graphs or images. There are two types of graphs in this Figure and it is not specified which one is used in each part. The B part is difficult to understand, as there is not clue what is the left or the right images, the lanes are not labeled and some molecular markers below 75 kDa are missed, so the reader can see what the authors mention in their manuscript about the MW of the band. Also, in the figure legend is missed the information about what type of samples are used, control? Patients?
Answer to comment 4: Thanks for this comment. We added more information in the legend of Supplementary Figure 1. In addition, the MWs of the various bands have been added.
Comment 5: Figure 2 only shows 1 spermatozoon. It is difficult to guess a clear pattern of immunofluorescence with only 1 cell.
Answer to comment 5: We see that this Figure has a very clear IF pattern. The same pattern is confirmed in Figure 4.
Comment 6: Authors state (lanes 306-307) that “an increase in the 306 levels of FSHR protein expression after 60 minutes of incubation in all patient samples”. However, this is not visualized at all in the Figure 3B. This referee cannot see any increase in any patient. In addition, the figures of western blot are so small and not clearly labeled their lanes that it cannot be properly visualized.
Answer to comment 6: The increase in the expression can be clearly visualized in panels D and E which, as the legend indicates, are the densitometric analyses of the FSHR/tubulin ratio in controls (panel D) and patients (panel E). Dark histograms express these levels in the samples before FSH incubation, and the white ones express these levels in the samples after FSH incubation. Patient samples (panel E) show an increase in expression after incubation compared to before incubation (the white histograms are higher than the dark ones in panel E).
Comment 7: Therefore, the histograms included in figure 3 seem not to be related to the western blot images. For instance, in D, the control 1 has more than 2-fold increase after FSH treatment; this is not seen in the related western blot in B, at least this referee visualizes not change o, if any, a decrease after FSH. This should be well explained.
Answer to comment 7: This is because the histograms indicate the FSHR/tubulin ratio (as written in the legend and in the figure itself) and not the FSHR. This adjustment was made not to analyze the absolute expression (which would have been a bias), but to standardize the levels based on the number of cells analyzed. Tubulin is expressed in all spermatozoa.
Comment 8: How can it be possible that expression levels of a protein can increase in spermatozoa? and even more challenging, after only 60 minutes?. This assumption is difficult to scientifically be understood (lanes 306-308).
Answer to comment 8: We understand your concern. We do not have an answer to this. Since the expression of FSHR in spermatozoa represents a relatively new finding, even its possible modulatory pattern is unknown. Recent evidence indicates that spermatozoa carry several transcripts and proteins, calling into question transcription silencing, which is thought to occur in spermatozoa. Consequently, a small amount of sperm RNA can be synthesized by de novo transcription in mature spermatozoa, as a small portion of the sperm genome remains packed by histones (doi: 10.1093/humupd/dmab034). This indicates that the molecular biology of spermatozoa still needs to be better elucidated and does not exclude the existence of a possible modulatory pattern of FSHR expression occurring in spermatozoa.
Comment 9: In legend of Figure 4 it is stated that “After incubation, it appeared in the post-acrosomal region, neck, mid-329 piece, and tail”, but the images this can only be seen in 1 patient, or maybe 2...
Answer to comment 9: The legend in Figure 4 clearly explains that this was found in Patient 4: “Magnification from Patient 4: before incubation, the protein was not identified (Left panel). After incubation, it appeared in the post-acrosomal region, neck, midpiece, and tail (Right panel) (B)”. We added red arrows in the magnified images (panel B, on the right).
Comment 10: The type of graphs in Figure 5 are not explained in the legend.
Answer to comment 10: These are box plots. A box plot is widely used in literature to report the distribution of values. It is particularly useful when data are not normally distributed since it allows us to identify the median value, the interquartile range, and the min and max values of the sample. A brief explanation is reported in the revised legend in Figure 5.
Comment 11: In lanes 355- 356 is mentioned “In in-355 toto semen, incubation with FSH reduced the percentage of spermatozoa with H-MMP 356 only in normozoospermic men at all concentrations tested”. However, in Table 2, this reduction although significant is so small. This is important because in other data of the table, there are bigger effects (semen pellet asthenozoospermia there is a reduction from 100 to 87,8 and is not mentioned. Besides, the seemed reduction in H-MMP mentioned by the authors in toto semen from normozoospermic men is not related with a significant increase in the low-MMP.
Answer to comment 11: In the article, we highlighted only the differences that reached statistical significance. A larger difference between two sample is meaningless unless it is significantly different. In fact, if no statistical difference is found, the samples cannot be defined as “different”. The significance is indicated with asterisk. This is a basic principle of statistical analysis and we are surprised to find such an observation.
Comment 12: Results about AKT phosphorylation are unclear and difficult to convince with the western blot presented. It is unwarranted any conclusion based on these images. It is really hard to visualize AKT band and even harder to see any effect.
Answer to comment 12: Panel A has now been enlarged to make the bands clearer. In the attached file, we have included an edited version of the Figure for the Reviewer, with blue arrows indicating the bands.
Comment 13: Authors are not convincing when explain the different molecular weight of the band obtained in their western blot. Panza et al (2020) previously showed that the MW of the FSH-R is found about 75 kDa.
Answer to comment 13: We see no other reasons to explain this result other than post-translational modification of the protein or the presence of a differently glycosylated form of FSHR in spermatozoa.
Comment 14: In order to explain their results authors used many hypothesis (Discussion), but convincing data are not provided nor newest data, apart from the already published by Panza et al., 2020.
Answer to comment 14: We do not agree with this Reviewer’s opinion, as explained in the answers to the comment 1.

Reviewer 2 Report
In this paper, it has been shown the expression of FSHR in different regions of human spermatozoa and in contrast with previous studies it is reported a negative effect of FSH on two important sperm function parameters. This study is well-designed and written. The flow chart is clear and self-describing and patient selection is appropriate. The figures are of good quality and convincing.
I have some comments:
This study deals with sperm functional parameters as motility and mitochondrial function. However, the authors do not provide a scientific background on sperm physiology. In the introduction, a brief scientific description of sperm physiological parameters and their role in fertilization competence and capability must be provided. This is important for the readership of IJMS which is not expert in reproductive physiology.
Line 97 authors must provide the rationale for investigating the spermatozoa after swim-up and a detailed description of the swim-up as the method used to obtain the best sperm population for ART techniques.
Methods. Authors must describe how they evaluated sperm motility. This is the most difficult parameter to be assessed and compared. So, in absence of a computerized system such as the CASA (computer-aided aided sperm analysis), it is necessary that motility must be evaluated always by the same expert operator. This aspect must be assessed and authors must be sure of the reproducibility of the results otherwise these may remain questionable.
In Figure 1 remove materials and methods from the legend and move them in the most appropriate section.
Figure 2 and 4 put arrowheads indicating the IF
Minor points replace CO2 with CO2
In conclusion, this study provides a very clear and detailed description of FSH and FSHR in spermatogenesis and sperm functionality which is of interest since it is not a well-known topic. The expression of FSHR in human spermatozoa represents an interesting novelty and this study adds a contribution to the knowledge of the effects that some substances may exert on spermatozoa either ejaculated or capacitated in order to ameliorate ART success and outcome.
This paper may be accepted after minor revision.
Author Response
Answers to the Reviewer #2 comments
Manuscript ID ijms-2223251
Comment 1: In this paper, it has been shown the expression of FSHR in different regions of human spermatozoa and in contrast with previous studies it is reported a negative effect of FSH on two important sperm function parameters. This study is well-designed and written. The flow chart is clear and self-describing and patient selection is appropriate. The figures are of good quality and convincing.
Answer to comment 1: We thank this Reviewer for taking the time to read and constructively comment on the article.
Comment 2. This study deals with sperm functional parameters as motility and mitochondrial function. However, the authors do not provide a scientific background on sperm physiology. In the introduction, a brief scientific description of sperm physiological parameters and their role in fertilization competence and capability must be provided. This is important for the readership of IJMS which is not expert in reproductive physiology.
Answer to comment 2: As requested by the Reviewer, a paragraph on sperm physiology, motility, and mitochondrial function has been added in lines 55-60.
Comment 3: Line 97 authors must provide the rationale for investigating the spermatozoa after swim-up and a detailed description of the swim-up as the method used to obtain the best sperm population for ART techniques.
Answer to comment 3: We appreciated this suggestion. The rationale for investigating swim-up, in-toto semen, and semen pellet is specified in lines 73-74 (the aim of the study). The swim-up technique is described in lines 127-133, where we have added a brief explanation of how swim-up is able to separate the highly motile spermatozoa.
Comment 4: Methods. Authors must describe how they evaluated sperm motility. This is the most difficult parameter to be assessed and compared. So, in absence of a computerized system such as the CASA (computer-aided aided sperm analysis), it is necessary that motility must be evaluated always by the same expert operator. This aspect must be assessed and authors must be sure of the reproducibility of the results otherwise these may remain questionable.
Answer to comment 4: To assess sperm motility we used the same criteria detailed in the WHO fifth manual (line 115). We have added section 2.5 to the revised version of the manuscript to explain how sperm motility was assessed. A well-trained and experienced Seminologist (N.B. author), with more than 20 years of experience in semen analysis, was involved in the assessment of sperm motility (please see lines 149-161).
Comment 5: In Figure 1 remove materials and methods from the legend and move them in the most appropriate section.
Answer to comment 5: Methodological details have now been removed from the legend of Figure 1 (please see lines 90-93). The same details are reported in section 2.5 of the revised version of the manuscript.
Comment 6: Figure 2 and 4 put arrowheads indicating the IF.
Answer to comment 6: Done, as requested (please see Figures 2 and 4).
Comment 7: Minor points replace CO2 with CO2.
Answer to comment 7: Done as requested (please see lines 130 and 144).
Comment 8: In conclusion, this study provides a very clear and detailed description of FSH and FSHR in spermatogenesis and sperm functionality which is of interest since it is not a well-known topic. The expression of FSHR in human spermatozoa represents an interesting novelty and this study adds a contribution to the knowledge of the effects that some substances may exert on spermatozoa either ejaculated or capacitated in order to ameliorate ART success and outcome. This paper may be accepted after minor revision.
Answer to comment 8: Thank you for your suggestions. We reviewed the article based on the comments you gave us.
Reviewer 3 Report
The authors aim to evaluate whether the follicle-stimulating hormone (FSH) receptor (FSHR) is expressed in human spermatozoa and the effects of FSH incubation on sperm function. The objective is interesting and could have practical applications. However there are some points that the authors should clarify. The major limitation is the number of samples. It seem very reduced for the number of analyzed performed by authors.
Experimental design (page 2 line 87 to 100):
This this not clear.
“accomplished by evaluating mRNA and protein expression in 4 consecutively enrolled patients” – Patients or controls? Why patients?
“the localization of FSHR protein expression by both WB” WB does not give the protein localization
“we repeated the latter experiments using the in-toto semen and the semen pellet” How? The same individual where analyzed by in toto an semen pellet and swim up? How the samples were divided?
Figure 1. It is missing the number of samples in each analysis
Treatment (page 4 line 158 to 163):
The same sample was incubated in all concentrations? The samples were divided? How was this organized?
Without this information clearly presented is impossible to correctly interpret the results presented.
2.7. Immunofluorescence analysis
Line 229. Why the treatment with RNAse?
Results
Is there any possible explanation for the difference in the mRNA expression between patient 3 and 4?
In protein data patient 4 does not seem to have the same level of protein amount as had in mRNA any justification?
Fig 2- seem unspecific signal
Scheme 2 The DAPI were the same among experiences? It seems a lot different

Author Response
Answers to the Reviewer #3 comments
Manuscript ID ijms-2223251
Comment 1: The authors aim to evaluate whether the follicle-stimulating hormone (FSH) receptor (FSHR) is expressed in human spermatozoa and the effects of FSH incubation on sperm function. The objective is interesting and could have practical applications. However there are some points that the authors should clarify. The major limitation is the number of samples. It appears very reduced for the number of analyzed performed by authors.
Answer to comment 1: We thank the Reviewer for taking the time to read and constructively comment on the article. We understand her/his point of view. However, the number of patients enrolled was sufficient to reach the statistical significance based on the results of the power analysis (please see section 2.10).
Comment 2. Experimental design (page 2 line 87 to 100): This this not clear.
- “accomplished by evaluating mRNA and protein expression in 4 consecutively enrolled patients” – Patients or controls? Why patients?
- “the localization of FSHR protein expression by both WB” WB does not give the protein localization
- “we repeated the latter experiments using the in-toto semen and the semen pellet” How? The same individual where analyzed by in toto an semen pellet and swim up? How the samples were divided?
Answer to comment 2: We appreciated this comment. Please find the point-by-point replies below.
- This section is merely descriptive and was undertaken to allow us to confirm or not the expression of FSHR. Based on the positive results of this first step, we started recruiting patients and controls (as detailed in Figure 1). Therefore, we have replaced the word “patients” with “subjects”. They were enrolled consecutively (the only enrolment criterion was the presence of spermatozoa in their ejaculates). Their sperm parameters are detailed following: two had normozoospermia, one oligo-asthenozoospermia, and one asthenozoospermia (please see lines 81-82 and 250-255).
Conventional sperm parameters of the subjects enrolled in this study
|
Sperm concentration (million/ml) |
Total sperm count (million/ejaculate) |
Progressive motility (%) |
Total motility (%) |
Normal forms (%) |
Leukocyte concentration (million/ml) |
1 |
65 |
292.5 |
32 |
61 |
14 |
0.65 |
2 |
67 |
167.5 |
32 |
50 |
5 |
1.34 |
3 |
2 |
6 |
16 |
60 |
7 |
0.12 |
4 |
110 |
385 |
25 |
75 |
10 |
0 |
l.l. |
>15 |
>39 |
>32 |
>40 |
>4 |
<1 |
l.l., lower limit according to the WHO criteria (2010)
- We reworded the sentence as follows: “Next, we evaluated the effects of hpFSH (0 and 30 mIU/mL) in swim-up separated human spermatozoa on FSHR protein expression and localization by WB and IF, respectively” (lines 85-86).
- Sample processing is detailed in section 2.3. In particular, the swim-up technique is described in lines 127-133. Aliquoting of in-toto semen and semen pellet is described in lines 136-138.
We treated 1 ml of the liquefied semen by swim-up technique. The remaining sample was divided into two parts. One was centrifuged (pellet) and the other was not centrifuged (semen in-toto). The three samples thus obtained were then incubated with FSH and motility and mitochondrial function were evaluated (as detailed in Figure 1).
Comment 3: Figure 1. It is missing the number of samples in each analysis.
Answer to comment 3: Done, as requested. Please see Figure 1 of the revised version of the manuscript.
Comment 4: Treatment (page 4 line 158 to 163):
- The same sample was incubated in all concentrations? The samples were divided? How was this organized? Without this information clearly presented is impossible to correctly interpret the results presented.
Answer to comment 4: After liquefaction and evaluation of concentration and motility, 1 ml of the sample underwent to swim-up. The remaining untreated fraction was divided into one aliquot that was centrifuged (pellet) and another one that was not centrifuged (in-toto semen). For each of these three specimens (swim-up, pellet, and in-toto semen), we prepared four different aliquots of 10x106 spermatozoa each. Each of these aliquots for each group was incubated with vehicle (FSH 0 mIU/ml), 1 mIU/ml FSH, 10 mIU/ml FSH, and 30 mIU/ml FSH. Motility was assessed before and after 60 minutes of incubation. Based on the sample volume, not all patient and control samples could be used for all experiments. The number of samples we were able to use for each experiment is shown in Figure 1. We have modified section 2.5 section in an attempt to enhance its clarity.
One aspect that we would like to underline is that a previous study (Panza et al., 2020) published on this topic only measured total motility. However, this parameter has limited value in clinical practice. Furthermore, the authors pooled samples from different patients to run the experiment s and this is a very important study drawback of the study. We have not pooled semen samples from patients or controls.
Comment 5: 2.7. Immunofluorescence analysis: Line 229. Why the treatment with RNAse?
Answer to comment 5: The nuclei were counterstained with DAPI, and to avoid a possible interaction of the dye with mRNAs that would have given a background also in the cytoplasm due to the presence of polyribosomes and the RER, we performed a pretreatment with RNAse, thus obtaining the identification of the focal plane, where the cells were located, with the blue nuclei containing empty nucleoli at this point (cleared of pre-mRNAs, or heterogeneous RNAs) and a perfectly dark cytoplasm.
Comment 6: Results
- Is there any possible explanation for the difference in the mRNA expression between patient 3 and 4?
- In protein data patient 4 does not seem to have the same level of protein amount as had in mRNA any justification?
Answer to comment 6:
- The conventional sperm parameters of the subjects enrolled in this part of the study are shown in Supplementary Table 1. Subjects 1 and 2 had normozoospermia, subject 3 had oligo-asthenozoospermia, and subject 4 had asthenozoospermia. The marked reduction in sperm count in subject 3 may account for the difference in results between subjects 3 and 4.
- In our experience, sometimes cells show some sort of negative feedback between mRNA and protein expression. In other words, low levels of protein can be associated with high levels of RNA, which will be translated into protein to restore protein levels. On the other hand, high protein levels can associate with low mRNA levels, thus leading to a reduction of protein levels. These mechanisms can be used by the cells to keep protein levels within a certain range.
Comment 7: Fig 2- seem unspecific signal
Answer to comment 7: Fluorescence, indicated by the red arrows, locates in the post-acrosomal region, midpiece, and flagellum. The specificity of the results is supported by the data shown in Supplementary Figure 1, where the positive control samples show the same fluorescence as shown in Figure 2, while the negative control samples show no fluorescence. Furthermore, Figure 4 largely confirms these findings.
Comment 8: Scheme 2 The DAPI were the same among experiences? It seems a lot different
Answer to comment 8: We used the same DAPI (Sigma-Aldrich, St. Louis, MO, USA) for all the experiments. Details are provided in line 214.
